# Implementation of a Single-Tooth Pre-Doctoral CAD/CAM Dentistry Curriculum at UIC: History, Description, SWOT Analysis, and Quantitative Evaluation

**Kristen Vlagos [1], Daisy Salazar [2], Andrea Reale [3], Cortino Sukotjo [3], Stephanie Clarke [3], Stephen D. Campbell [3] and Alexandra E. Rodriguez [3,***

[1] Edward Hines, Jr. VA Hospital, Hines, IL 60141, USA; Kristen.Vlagos@va.gov
[2] Eastman Institute of Oral Health, University of Rochester, Rochester, NY 14642 , USA; daisy_salazar@urmc.rochester.edu
[3] Department of Restorative Dentistry, University of Illinois at Chicago, Chicago, IL 60607, USA; areale@uic.edu (A.R.); csukotjo@uic.edu (C.S.); clarkes@uic.edu (S.C.); stephend@uic.edu (S.D.C.)
*** Correspondence: alerod@uic.edu

**Abstract:** A comprehensive CAD/CAM dentistry curriculum that includes broad aspects of single tooth restoration has been implemented at the University of Illinois College of Dentistry (COD) since 2010. The purpose of the program is to promote an educational environment that utilizes current technology to enhance the quality and efficiency of single-tooth dental treatment offered to patients, while also preparing graduating students to apply recent and future clinical advances. This article describes the history, organization, rationale, and objectives of the tooth-supported restorative aspects of our comprehensive pre-clinical and clinical CAD/CAM curriculum and presents the educational and clinical outcomes of this program.

**Keywords:** predoctoral; CAD/CAM; curriculum; SWOT analysis

## 1. Introduction

The prevalence of intraoral scanning systems has grown substantially in recent years due to technological advances, resulting in improved accuracy/precision and efficiency while reducing user barriers such as high ownership costs and steep learning curves [1]. It is estimated that more than 20% of general dentists have intra-oral in private practice [2]. Due to both economic and patient demand, the preferred fabrication method of dental restorations is expected to continue its shift towards computer-aided design/computer-aided manufacturing (CAD/CAM) [3,4]

CAD/CAM restorations offer the potential for single visit treatment, low cost, high accuracy, high strength, and esthetics, qualities that are difficult to achieve through traditional methods [5–9]. Until recently, CAD/CAM innovation has been mostly embraced by private dental clinicians while dental education had been relatively slow to adapt [10]. With emerging evidence of the accuracy, predictability, and efficiency of digitally designed and fabricated restorations there has been an increase in the number of US dental schools (over 65%) that have implemented CAD/CAM technologies into the curriculum [11].

In 2016, the American College of Prosthodontics (ACP) launched a digital dentistry initiative that resulted in the development of a comprehensive curriculum framework that included competency statements, learning objectives, implementation strategies, and extensive resources to support schools [12–15]. The UIC CAD/CAM dentistry experiences served as the core of the ACP's Digital Dentistry Curriculum and resources [12]. On July 2019 the Committee on Dental Accreditation (CODA) put forth new accreditation standards that mandate all pre-doctoral educational programs to incorporate emerging didactic and clinical technologies in their educational programs [8]. Of the dental schools that have put a digital curriculum into effect, it is unclear how these various programs are characterized

and what their outcomes have been. Only a few authors have described the process of implanting CAD/CAM in a dental school curriculum and described the outcome of doing so. Literatures supports that students will be more likely to prefer and adopt technology in their future practice when it has a perceived value for improving efficiency [16,17].

At the University of Illinois at Chicago, College of Dentistry (UIC-COD) a single-tooth digital dentistry educational program was initiated in 2010 with the goal of training students to become competent in providing CAD/CAM tooth-supported restorations. The digital dentistry program at UIC-COD is intended to increase student exposure to current CAD/CAM restorative methods and emphasize them as tools for providing minimally invasive dental treatment while improving dental education and patient experience. The aims of this article are to: (1) describe the implementation of the CAD/CAM dentistry curriculum at the UIC-COD; (2) describe the outcomes of the curriculum implementation by using a SWOT analysis; and (3) describe the clinical experiences/production of this program for the first three years of clinical implementation.

## 2. Single-Tooth Digital Curriculum Overview

In 2010 a digital initiative began at UIC-COD with the opening of the Center of Digital Excellence (CDE). The available technology was carefully assessed and decisions were made in regard to specific scanning, design, and milling platforms. With a wide range of products available on the market, the most important factors affecting the Restorative Department's decision included portability, open software interface, robust design, and production capabilities, as well as documented evidence of efficacy.

A core group of 12 faculty members was identified and trained to support the CAD/CAM dentistry educational initiative, followed by the gradual development and introduction of content into our pre-patient care courses in 2013. This included the development of a digital philosophy document, competency statements, and detailed learning objectives to clearly define the student learning expectations.

The single-tooth digital program is divided into two branches: a pre-clinical and a clinical component. Initially, CAD/CAM training was incorporated into the pre-patientcare fixed prosthodontic course. However, in order to adequately prepare students to use digital scanning, design, and fabrication in the clinic, it was decided that a stand-alone digital course was warranted. A digital learning lab facility was completed in 2015 in support of the program.

The student clinical experiences were initiated in 2016 as part of a dedicated Digital Dentistry Clinic and included the hiring and training of a digital designer/technician to support the learning and clinical care. The CDE created the impetus to formalize a digital curriculum and lay out a plan for future growth. A comprehensive and detailed financial analysis and budget-planning process as it related to the ongoing operational expenses associated with the digital education and patient care programs was completed in 2016. It demonstrated our ability to reduce the expenses associated with tooth supported inlay/onlay/crown restorations in the predoctoral clinics by 40% [7].

The program has continued to develop and expand over the years, with the addition of more trained faculty, the introduction of complete digital workflows into the pre-doctoral implant curriculum, and the establishment of an advanced digital pilot program for fourth year students. A summary of the history of the implementation of the Digital Dentistry curriculum is shown in Figure 1.

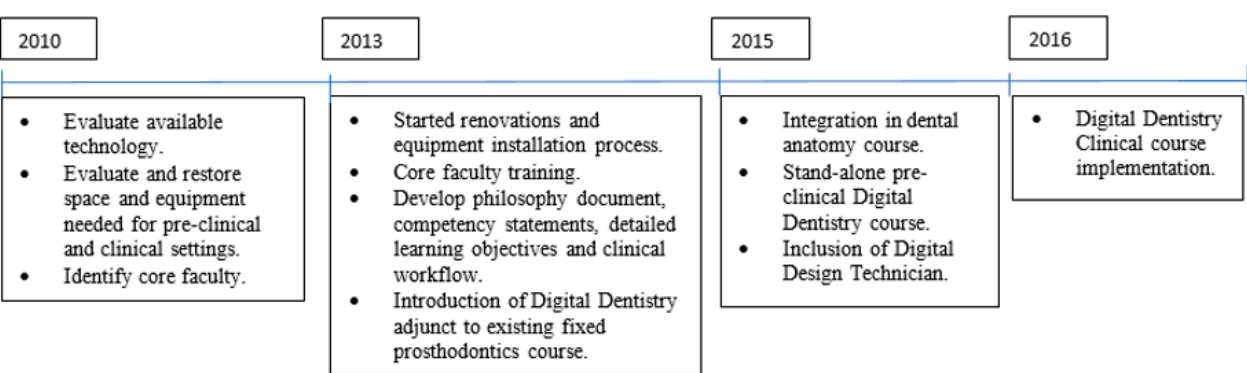

**Figure 1.** History of Implementation of CAD/CAM curriculum at UIC.

### 3. Pre-Clinical Component

Students are introduced to intraoral scanning during the first semester of dental school. As part of the dental anatomy course, students digitize a wax-up with the use of the E4D PlanScan intraoral scanner (Planmeca E4D Technologies, Richardson, TX, USA), and then compare their digital reproduction to an ideal tooth form in an evaluative software program (Planmeca E4D Technologies and E4D Compare, Richardson, TX, USA). This initial scanning exercise facilitates self-assessment and acclimatizes students to interpretation of 3D computer modeling.

This skill is further refined during the comprehensive pre-patient care digital dentistry course taken during the fall of the second year DMD program. This 16-week course focuses on single-tooth restorations and includes both didactic and hands-on pre-clinical sessions. There are roughly 70 pre-doctoral students and 52 students from the advance-standing program (an accelerated DMD curriculum for internationally trained dentists) that rotate through several clinical and lab exercises accounting for a total of about 33 hours of hands-on training. Substantial clinical instruction time is made possible because most didactic lessons are geared toward self-guided learning utilizing an online education portal (Blackboard Inc., Washington, DC, USA). Content is reinforced through classroom review and material comprehension gauged through weekly quizzes. These brief assessments are low stakes to allow focus to remain on clinical time, yet they allow the faculty to identify any possible learning gaps and address them more readily. The course primarily concentrates on small group learning conducted in a dedicated digital design lab that encourages active participation. The lab accommodates up to 18 students with 11 computer and scanner workstations. With two large, 84 monitor displays, a faculty leader guides students as they follow along in real time through the scanning and designing workflow. Additional floating faculty are present for individualized student attention, providing a faculty to student ratio of 1:6. In total, students complete at least two single-tooth digital designed and milled restorations following the prescribed workflow in its entirety. They perform additional tooth preparations and scan/design exercises as well as a "scan on partner" experience, gaining a more clinical feeling of intraoral scanning. Student progress is also assessed through daily instructor feedback and modular examinations Figure 2.

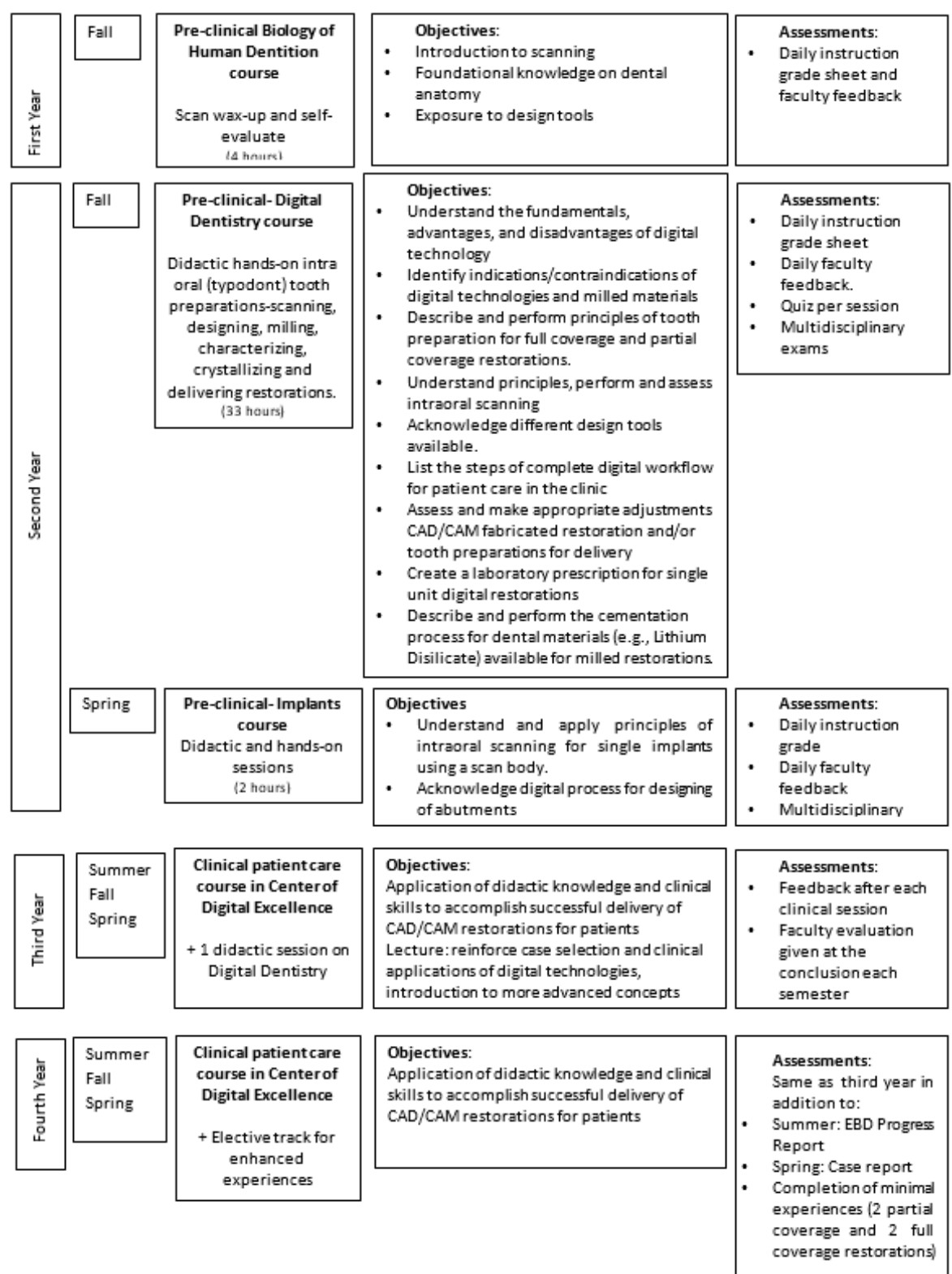

**Figure 2.** Distribution of CAD/CAM dentistry in the UIC curriculum.

## 4. Clinical Component

Direct patient care begins when the students matriculate into the third year. This translates into approximately 244 students rotating the CDE. Over the course of two years in clinical rotations, students are expected to complete at least two partial coverage (either inlay and/or onlay) and two full coverage (crown) digitally fabricated restorations.

These minimal essential experiences were introduced in 2018, following a two-year pilot program in which student production was observed and a consensus on a reasonable digital expectation was determined.

All digital restorative treatment for UG dental patients is completed in a dedicated digital clinic (CDE) that is separate from the COD's five other comprehensive care clinics. The intraoral scanner used is Trios 3 (3Shape, Copenhagen, Denmark). The CDE is located in close proximity to the digital design lab and milling center. A digital lab technician supports the in-house CAD/CAM restorations workflow of all the UGs and PGs. The milling center has three PlanMill 40 (Planmeca E4D Technologies, Richardson, TX, USA) milling machines. From a logistical standpoint, this well-controlled, centralized environment greatly enhances clinical efficiency and allows for the close monitoring and maintenance of equipment and supplies. Since the inauguration of the clinic, appointment availability has increased as necessitated. Currently, the clinic is open five days per week, with approximately six to eight chairs, and is supervised by up to two faculty members per day. All assigned digital dentistry faculty have advanced restorative training and are specifically calibrated to guide the students through the CAD/CAM process to ensure protocols are uniformly followed.

Patient care initiation begins with the identification of patients that may benefit from an indirect restoration Figure 3. A case selection checklist is used to ensure standardization among the students and faculty. Subsequent to the initial or periodic examination, the student and supervising faculty develop a comprehensive treatment plan that incorporates any digitally fabricated restorations. Before the plan is finalized, the student may request a consult with a digital faculty member to confirm appropriateness for the patient. To differentiate the in-house CAD/CAM fabricated from outsourced conventional indirect laboratory restorations, and more accurately track experiences, a digital dental code is entered into the electronic patient record (AxiUm). This is done by adding modifiers to the CDT code base, to clearly indicate the digital workflow. The faculty and Managing Partners of the Comprehensive Care Clinics are calibrated through regular faculty development sessions, where patient selection, coding, fees, and workflow are discussed.

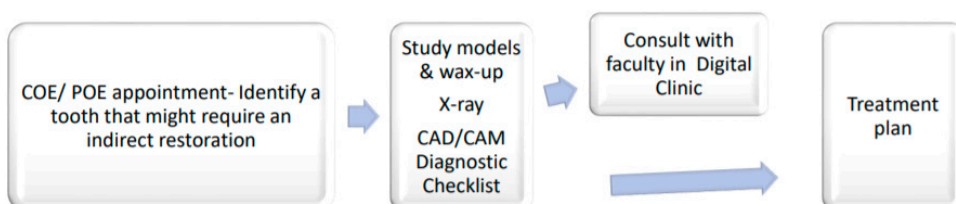

**Figure 3.** Clinical workflow for in-house CAD/CAM patient treatment planning. COE = comprehensive oral exam; POE = periodic oral exam; CDE = Center of Digital Excellence.

Clinical procedures are usually completed in two sessions. The first session encompasses preparation and scanning while the second session includes the try-in, crystallization/glazing, and delivery Figure 4 of the lithium disilicate restoration. While single visit procedures are achievable, they are not expected as most procedures are completed in two appointments. To help students prepare for clinical activities, resources are available through the Blackboard site (Blackboard Inc., Washington, DC, USA) and include the course syllabus, workflows, intraoral scanning workflow, scanner user manuals, preparation guidelines, and various material handling instructions. In an effort to optimize learning and procedural efficiency, the digital technician is an important part of the workflow, allowing students to focus on direct patient care. Design collaboration and communication between student and technician is an expectation. In this centralized production model, the technician fulfills all digital designs with final review and approval by faculty for quality assurance.

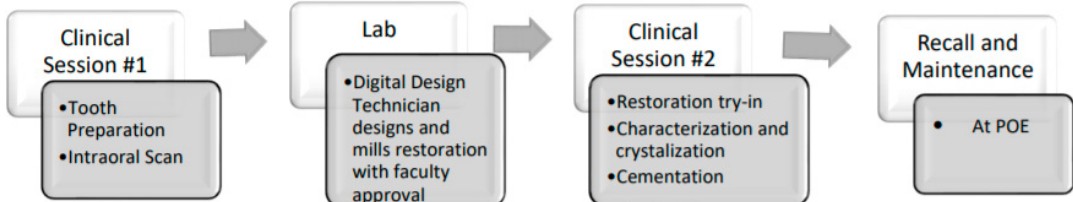

**Figure 4.** Predoctoral clinical and laboratory single-tooth digital workflow for preparation, fabrication, delivery, and recall of tooth-supported restorations. POE = periodic oral exam.

Students are given daily formative feedback on their clinical performance by attending faculty. Summative assessments are also shared each semester and include consideration of other factors such as professionalism, communication, diagnostic abilities, and application of clinical knowledge. In the first semester of their fourth year, students are required to submit an EBD progress report stating the number of restorations completed and comparing a digital vs. conventional impression experience. This report allows students to complete an analytical literature review, encouraging them to critically reflect on emerging digital trends, studies, and technologies. Faculty have the opportunity to identify students who may be deficient in digital experiences and provide them with additional assistance in selecting appropriate cases. Prior to degree completion, students are required to submit a patient case report that documents one of their patients. This assignment not only summarizes the treatment they provided, but also asks them to reflect on the outcome and their clinical performance. Descriptions of all clinical assessments for both D3 and D4 students are summarized in Figure 2.

## 5. Single-Tooth Digital Dentistry Program SWOT Analysis

To better understand the program's overall fitness and sustainability, a SWOT analysis was conducted Table 1. This assessment method is simple yet thorough and helps to identify internal and external factors affecting the maintenance and growth of this educational model.

**Table 1.** SWOT analysis for pre-doctoral single-tooth digital program.

| Strengths | Weakness |
|---|---|
| 1. Immediate feedback for self-assessment<br>2. High student-teacher contact time<br>3. Integrated learning environment<br>4. Improved communication with lab<br>5. Enhanced patient experience | 1. Reduced digital design requirement in clinic<br>2. More challenging for students with weaker technical abilities<br>3. Need support from digital technician and IT infrastructure<br>4. Administrative burden |
| **Opportunities** | **Threats** |
| 1. Increase student productivity<br>2. Institutional cost savings<br>3. Increase access to care-cost and treatment time reduction<br>4. Expansion into other clinical areas | 1. Maintenance needs-software upgrades, equipment maintenance/replacement, data storage and IT infrastructure<br>2. Need for ongoing faculty professional development<br>3. Allocated funding or budgeting<br>4. Need for support from the higher administration |

### 5.1. Strength

A subjective assessment of the digital program identifies several key strengths. It is clear that the educational value of the single-tooth digital program far exceeds the explicit purpose of teaching students a novel technology. Intra-oral scanning provides students

with immediate feedback on the quality of their tooth preparations and allows them to more readily improve their clinical performance in real time. This opportunity to self-assess and correct helps students build on their self-awareness, confidence, and critical thinking abilities. The significant amount of individualized faculty feedback provided throughout the digital curriculum is also notable. By investing in high quality learning environments with robust faculty to student ratios, more meaningful student-teacher learning partnerships are established and engagement is maximized. Further, our intensive and comprehensive single-tooth digital curriculum challenges students to synthesize knowledge related to a variety of disciplines such as dental anatomy, occlusion, and material science. The direct line of communication with the in-house lab also provides an extra measure of quality assurance with complete oversight from start to finish. Lastly, the digital program has proven to be extremely positive for our patient population with the more timely completion of care and increased comfort and access to more conservative restorative options.

*5.2. Weakness*

A drawback of our current digital program is its limited designing opportunity for students. Educational focus is placed on scanning and restoration delivery, allowing an emphasis on developing those competencies and a realistic simulation for the private practice setting and minimizing the clinical burden on students. However, by skipping mandated clinical design, students may miss the opportunity to fully evaluate their preparation and appreciate its effect on the final outcome. Some may feel they are not being provided with enough design instruction for private practice and may need to seek out additional continuing education after graduation. There is also concern about how the toll of operating in a highly technical environment that can affect student, faculty, and staff alike. To offset these concerns, design collaboration and communication between the student and technician is an expectation, thereby providing more efficient learning opportunities. An elective track course has also been implemented for fourth year students who would like to expand their knowledge and skills.

*5.3. Opportunity*

The implementation of a digital system offers the opportunity for our institution to reduce its economic burden and improve its clinical management. The reduction of disposable materials, storage space for working casts, and the time and personnel required to pack and ship lab work all equate to cost savings. While we offer reduced fees for some digital treatment, an expansion of the programs could further lower treatment costs and may allow patients to receive needed treatment they might otherwise decline. With fewer appointments and shorter treatment times, it is expected that the College will be able to accommodate more patient visits per year, thus increasing clinical revenue.

The expansion of digital scanning into other clinical areas such as removable prosthesis, occlusal guards, orthodontic appliance, surgical guides, etc., has the potential to further reduce costs and increase revenues and access to care. Because of this improved efficiency, students will have more time to expand their educational opportunities in research, service projects, or community outreach.

*5.4. Threat*

The ongoing maintenance needs required to support the continually evolving technology represent a persistent and ongoing threat to the digital program. Software upgrades, equipment maintenance/replacement, data storage solutions, and IT infrastructure optimization are hurdles we must address. The continued growth of the program may also require the hiring of additional staff or auxiliary personnel to help distribute increasing administrative duties. In addition, there are concerns regarding the provision of adequate ongoing faculty and staff development to ensure best educational and patient care experiences. Ultimately, all of these demands rely on stable and long-term financial and ad-

ministrative support. Allocating sufficient support can be challenging within the confines of complex university resource planning. Long-term and big-picture thinking is essential for the program to succeed in its mission to advance dental education and patient care.

## 6. Clinical Trends

To investigate the clinical trends resulting from the integration of digital dentistry into restorative treatment, a record review of all patients treated with direct and indirect restorations was conducted for five academic years, from 15 August 2014 to 14 August 2019, including the first three years of implementation of the clinical CAD/CAM dentistry curriculum (Fig. 5). The University of Illinois at Chicago's Institutional Review Board approved the study (Protocol # 2020-1015). The restorations delivered: posterior amalgam 3–4 surfaces (PA), posterior composite 3–4 surfaces (PC), non-digital inlay and onlay metallic/ceramic (NDIOMC), non-digital porcelain fused to metal crown (NDPFM), non-digital all ceramic crown (NDCC), all ceramic crown digital (DCC), and inlay and onlay ceramic digital (DCIO) were recorded, calculated, and the trend over the years plotted Figure 5.

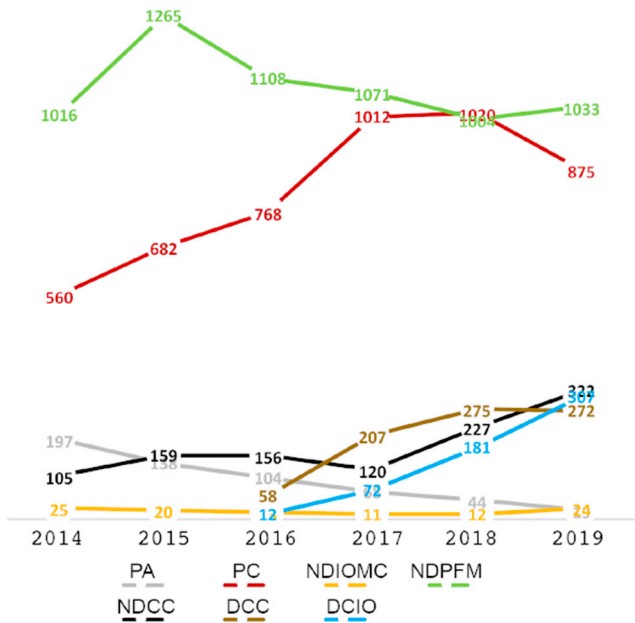

**Figure 5.** Total number of direct (amalgam and composite) and indirect partial and full coverage (conventional and digital) restorations delivered per year by predoctoral dental students from 1 June 2014 to 31 August 2019. Posterior Amalgam 3-4 Surfaces (PA), Posterior Composite 3-4 Surfaces (PC), Non-digital Inlay and Onlay Metallic/Ceramic (NDCIO), Non-Digital Porcelain Fused to Metal (NDPFM), Non-Digital All Ceramic Crown (NDCC), Digital All Ceramic Crown (DCC) and Digital Inlay and Onlay Ceramic (DCIO). NDCC was not digitally impressed in the college. DCC and DIO were digitally impressed and milled in the college.

Results show an emerging trend of increased tooth colored restorative materials, ceramic and composite, with a corresponding decreased use of traditional metallic materials, PFM, and amalgam. During the study period, we observed a steady rise in the number of digitally fabricated restorations for all types (partial and full coverage). The use of digital all ceramic (DCC) crowns has more than quadrupled over the last four years since the introduction of the single-tooth digital curriculum. At the clinic's onset, the total number of delivered units of digital inlay/onlay and crowns was 12 and 58, respectively. By 2019, the total number of inlay/onlay units grew to 367 and for crown units 272.

The number of PFM restorations appeared to have the most variability from year to year. While the use of PFM crowns demonstrated a 25% increase from 2014 to 2015, there was a subsequent decrease of 21% over the next three years. There was a slight 2% increase

in PFMs again from 2018 to 2019. Despite these fluctuations, PFM crowns remain a more frequently used material for full coverage restorations than all ceramic, either digital or conventional, as related to the college's philosophy.

The data revealed posterior composite is the direct restorative material of choice in the pre-doctoral clinic. Beginning in 2014, the placement of amalgam alloys was relatively low, with 197 total units compared to 560 units of composite completed annually. Data showed a steady decline in amalgam use, with a total reduction of 85% over the 6-year observation period. Conversely, there was a striking 80% increase in posterior composite restoration restorations between 2014 and 2017. However, after 2017 the rate of composite resin usage moderated with an increase of 0.8% in 2018 (1020). Posterior composites dramatically decreased by 14% to 875 in 2019. This was offset by an increase in ceramic partial and full coverage restorations, largely driven by the digital processes.

Along with digital restorations and composites, a strong growth trend was also observed for conventional all ceramic crowns. Conventional all ceramic crowns grew from 105 units in 2014 to 322 units in 2019. Despite the surge in non-digital crowns, the use of conventional lab fabricated inlay/onlay (NDIOMC) remained relatively steady, with no significant change in production.

## 7. Discussion

Dentistry is experiencing a rapid digital evolution. While private dental practice and industry have been the main drivers in the past, dental institutions and new generation dentists are now emerging as the new thought leaders [11–17]. Some authors have expressed how useful of a learning tool it is to have CAD/CAM systems in dental education and the great clinical outcomes that can be achieved. They also explain how CAD/CAM dentistry has been integrated into their dental school curriculums, with some similarities to the UIC implementation: evaluating the available technology and selecting the most appropriate one, then integrating the equipment into different pre-clinical didactic and hands-on courses, such as dental anatomy, fixed prosthodontics, esthetic dentistry, and using the equipment in a selected clinical area for CAD/CAM dentistry [18–20].

Drs. Browning, Harrison, and Reifeis, in their articles, reveal several areas where success and growth have been achieved by utilizing CAD/CAM systems in a dental school setting, similar to those described in this article in the strengths though our SWOT analysis [5,18,20]. The educational and clinical benefits of single-tooth digital workflows affirm this as a technology with vast future applications for dental instruction and patient care.

The SWOT analysis of UIC's digital curriculum brings recognition of the difficulties and threads in order that readers can take these into consideration when implementing CAD/CAM dentistry in a large dental institution, as also described by Jahangiri et al [10]. There is a need to recognize the importance of having ongoing IT support and infrastructure to grow in this technological era.

There were several limitations recognized in this study. It should be acknowledged that a reduced fee schedule was put into effect to make the digital partial coverage restorations more comparable with the large direct composite fees. This was done to support the best clinical decision making for patients. In addition, enforcing a minimum patient experience for students likely caused a substantial increase in digital procedures.

Observations of our clinical production over the course of five years reveal the growing use of large composite and ceramic restorations with a concurrent decline of amalgams and PFM crowns. This study provides a useful snapshot of a broader global trend in the types of materials and restorations used, as well as the impact of digitally designed and fabricated restorations [21]. While the application of indirect partial coverage restorations increased within our institution, a preference for direct composite restorations and full coverage crowns remains. The lower use of inlays and onlays, especially for novice practitioners, may be related to the learning curve, increased technical difficulty, and chair-time investment of performing these restorations [22]. Likewise, it may be that the vast majority of clinical

faculty covering the comprehensive clinics are continuing to teach what they have been doing and are simply less comfortable with digital workflows and materials.

In addition to preparing students with the knowledge and skills to enter into an era of rapidly changing technology and clinical practice, the use of digital technologies serves to reinforce a minimally invasive treatment approach for these new practitioners. Future long term studies are needed to assess if utilizing CAD/CAM restorations, especially those that are minimally invasive (i.e., inlay and onlay), will improve long-term outcomes, resulting in fewer root canal treatments and increased longevity of restorations, and providing higher tooth survival rates. We also need to assess if formal digital education and the development of competence will influence the student's future practice of dentistry. Awareness of how digital educational programs are enhancing student learning will help institutions improve their educational strategies and better prepare students to enter the private practice environment.

## 8. Conclusions

Implementing a digital curriculum has enhanced student education and patient care at UIC. It is expected that utilization of CAD/CAM restorations will continue to grow in the clinics as the technology improves and familiarity broadens amongst faculty and students. However, more time will be needed to determine if access to CAD/CAM resources will promote lasting changes to treatment planning philosophy and pedagogy.

**Author Contributions:** Investigation, A.E.R., D.S., A.R., S.C. and S.D.C.; Writing—original draft, K.V.; Writing—review & editing, C.S. and A.E.R. All authors have read and agreed to the published version of the manuscript.

**Funding:** This research received no external funding.

**Institutional Review Board Statement:** The University of Illinois at Chicago's Institutional Review Board approved the study (Protocol # 2020-1015).

**Informed Consent Statement:** Not applicable.

**Conflicts of Interest:** The authors declare no conflict of interest.

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
