# Peer review of "Implementation of a Single-Tooth Pre-Doctoral CAD/CAM Dentistry Curriculum at UIC: History, Description, SWOT Analysis, and Quantitative Evaluation"

_education, doi:10.3390/educsci11060301_

Round 1
Reviewer 1 Report
The article is interesting and well written but some minor point should be clarified. In the introduction, the line 33/36 are not clear. In the introduction it could be interesting for the readers to have some data that compare the digital dentistry procedures and the "analogic" one in therms of precision, accuracy, time and cost reduction.Author Response
Reviewer #1:
The article is interesting and well written but some minor point should be clarified. In the introduction, the line 33/36 are not clear. In the introduction it could be interesting for the readers to have some data that compare the digital dentistry procedures and the "analogic" one in therms of precision, accuracy, time and cost reduction.
Authors: Thank you very much for your valuable input. We have modified the introduction according to your recommendation.
Reviewer 2 Report
Dear authors,
I realize that authors have many journals to consider when they want to publish their work, so I appreciate your interest in Education Sciences; I am very sorry not to be able to write in a more positive way.
I agree with authors’ idea which is to educate the digital dentistry in recent years. Authors work on the education for the students in UIC-COD incorporating CAD/CAM curriculum. Table 1 is very suitable and easy to understand the CAD/CAM dentistry curriculum to readers because each curriculum clearly include the objectives to learn CAD/CAM restorations for patients. Furthermore, the workflows in Figure 1 and 2 help to understand the clinical step. However, the history is too long. Authors should summarize the flow of history in the diagram. Then, the first aim should be reconsidered.
SWOT analysis was used in this study, but the results from analysis was not clearly described. Since authors showed “evaluate using a SWOT analysis”, reviewer believe that “evaluate” include the showing results. Furthermore, “evaluate” does not applies to the aim because “evaluate” is a method to derive the results. So, authors should reconsider the second aim.
Authors should more describe the effect of education for CAD/CAM curriculum in discussion.
Authors should unify the words ”intra-oral” or “intraoral”.
I cannot find Table 2.
The abbreviations of Posterior Amalgam and Posterior Composite are different between text and legend in Figure 3.
Thanks for the opportunity to review this interesting paper.
Sincerely yours,
Author Response
Reviewer #2
I realize that authors have many journals to consider when they want to publish their work, so I appreciate your interest in Education Sciences; I am very sorry not to be able to write in a more positive way.
I agree with authors’ idea which is to educate the digital dentistry in recent years. Authors work on the education for the students in UIC-COD incorporating CAD/CAM curriculum. Table 1 is very suitable and easy to understand the CAD/CAM dentistry curriculum to readers because each curriculum clearly include the objectives to learn CAD/CAM restorations for patients. Furthermore, the workflows in Figure 1 and 2 help to understand the clinical step. However, the history is too long. Authors should summarize the flow of history in the diagram. Then, the first aim should be reconsidered.
Authors: thank you very much, we have created another figure (Figure 1) that summarize the history and we have modified the text accordingly.
SWOT analysis was used in this study, but the results from analysis was not clearly described. Since authors showed “evaluate using a SWOT analysis”, reviewer believe that “evaluate” include the showing results. Furthermore, “evaluate” does not applies to the aim because “evaluate” is a method to derive the results. So, authors should reconsider the second aim.
Authors should more describe the effect of education for CAD/CAM curriculum in discussion.
Authors should unify the words ”intra-oral” or “intraoral”.
I cannot find Table 2.
The abbreviations of Posterior Amalgam and Posterior Composite are different between text and legend in Figure 3.
Thanks for the opportunity to review this interesting paper.
Authors: Thank you very much for your kind and thoughtful input. We apologize that we did not include table 2 when we submitted the first paper. For the revised paper, we have included table 2 (swot analysis). We have modified all the abbreviations and some terminology accordingly. We have modified the discussion as you recommended as well.
Reviewer 3 Report
- Very few authors cited in the introduction!
- Insufficient volume of the whole article for the scientific level of the journal!
- Insufficient number of examples and unconvincing presentation of the positives of the program.
Author Response
Reviewer #3
- Very few authors cited in the introduction!
- Insufficient volume of the whole article for the scientific level of the journal!
- Insufficient number of examples and unconvincing presentation of the positives of the program.
Authors: Thank you for your valuable input. We have added another 9 references and we have added more discussion. We strongly believe that this paper is valuable for the readers since the implementation of digital dentistry curriculum is becoming a very hot topic among dental educators. Only limited papers available/published regarding this topic (curriculum and implementation).
Round 2
Reviewer 2 Report
Dear authors,
Thanks for the opportunity to review this interesting paper.
I don't have any comments.
Sincerely yours,
Reviewer 3 Report
The authors have tried to correct the article according to the given recommendations!